# B Cells Induce Early-Onset Maternal Inflammation to Protect against LPS-Induced Fetal Rejection

**DOI:** 10.3390/ijms242216091

**Published:** 2023-11-08

**Authors:** Gina Marie Uehre, Svetlana Tchaikovski, Atanas Ignatov, Ana Claudia Zenclussen, Mandy Busse

**Affiliations:** 1Experimental Obstetrics and Gynecology, Medical Faculty, Otto-von-Guericke University, 39108 Magdeburg, Germany; gina.uehre@med.ovgu.de; 2University Hospital for Obstetrics and Gynecology, Medical Faculty, Otto-von-Guericke University, 39108 Magdeburg, Germany; svetlana.tchaikovski@med.ovgu.de (S.T.); atanas.ignatov@med.ovgu.de (A.I.); 3Department of Environmental Immunology, Helmholtz Centre for Environmental Research-UFZ, 04318 Leipzig, Germany; ana.zenclussen@ufz.de; 4Saxonian Incubator for Translation Research, Leipzig University, 04103 Leipzig, Germany

**Keywords:** preterm birth, pregnancy, lipopolysaccharide, inflammation, B cells

## Abstract

The maternal balance between B regulatory (Breg) cells and inflammatory B cells is of central importance for protection against preterm birth (PTB). However, the impact of B cell signaling in early maternal and fetal immune responses on inflammatory insults remains underinvestigated. To understand which role B cells and B-cell-specific signaling play in the pathogenesis of PTB, the later was induced by an injection of LPS in B cell-sufficient WT mice, CD19^−/−^, BMyD88^−/−^ and µMT murine dams at gestational day 16 (gd 16). WT dams developed a strong inflammatory response in their peritoneal cavity (PC), with an increased infiltration of granulocytes and enhanced IL-6, TNF-α, IL-17 and MCP-1 levels. However, they demonstrated a reduced NOS2 expression of PC macrophages 4 h after the LPS injection. Simultaneously, LPS-challenged WT dams upregulated pregnancy-protective factors like IL-10 and TARC. The concentrations of inflammatory mediators in the placental supernatants, amniotic fluids, fetal serums and gestational tissues were lower in LPS-challenged WT dams compared to CD19^−/−^, BMyD88^−/−^ and µMT dams, thereby protecting WT fetuses from being born preterm. B cell deficiency, or the loss of B-cell-specific CD19 or MyD88 expression, resulted in an early shift from immune regulation towards inflammation at the fetomaternal interface and fetuses, resulting in PTB.

## 1. Introduction

During pregnancy, the maternal immune system has to protect the mother and unborn from the consequences of microbial exposure. Pathogen recognition is mediated by molecules called pattern recognition receptors—among them, a family of toll-like receptors (TLRs). TLRs are expressed by innate and adaptive immune cells, but their mRNA is also present in several organs such as the liver, lung, gut, ovary, placenta and brain [1].

TLR4 is important for the recognition of lipopolysaccharides (LPSs), immune-stimulatory components of the cell membrane of Gram-negative bacteria such as Escherichia coli (*E. coli*) [2]. Following the ligation of LPSs to TLR4, the adapter molecule MyD88 is recruited. This results in the activation of MyD88-dependent signaling pathways such as the c-Jun N-terminal kinase (JNK)/p38 mitogen-activated protein (MAP) kinase family and the transcription factor NF-κB. Additionally, LPS-TLR4 activates the TIR (toll/interleukin-1 receptor) domain-containing adaptor protein, inducing IFN-β (TRIF)-dependent/MyD88-independent pathways [3]. Both pathways lead to the production of several pro-inflammatory cytokines and chemokines. While it was shown that the production of IL-6 and IL-1β, among others, is mediated by MyD88-dependent signaling, TRIF-dependent signaling activates the expression of the chemokine RANTES (regulated on activation, normal T-cell-expressed and secreted) [4,5]. 

B cells express TLRs influencing their development, activation and differentiation when activated [6]. Besides TLR4, B cells also express another TLR involved in the LPS recognition, RP105 (CD180) [7]. It was shown that CD19 regulates signal transduction through RP105 [8]. LPSs activate B cells, increases their expression of co-stimulatory molecules, thereby enhancing their abilities as professional antigen-presenting cells [9] and drives their differentiation into plasma cells and B regulatory (Breg) cells [10,11].

How the B-cell response to maternal infection influences the outcome of pregnancy still needs further investigation. In our former work, we found an imbalance between pro- and anti-inflammatory B cells with a decreased frequency of IL-10-secreting Breg cells in women with preterm birth (PTB) [12,13,14]. Further, we showed the importance of functional B-cell-specific MyD88, IL-10 and CD19 signaling in mouse models of inflammation-induced intrauterine fetal death (IUFD) and PTB using intraperitoneal injection of LPS [15,16,17]. We also identified a role of B-cell-specific IL-10, CD19 and MyD88 in the appropriate intrauterine fetal development [18]. However, the exact mechanisms involving B cell signaling (MyD88, CD19) that underlie early maternal and fetal responses towards LPS challenge remain to be investigated. Therefore, to determine early immune signaling pathways activated by B cells, we analyzed the expression of maternal and fetal molecules, in particular cytokines and chemokines, 4 h after LPS injection in the present study.

## 2. Results

### 2.1. WT Dams Induce a Robust Immune Response Detectable in Peritoneal Lavage (PL) and Maternal Serum 4 h after LPS Challenge

To examine the early immune response to LPS challenge in gd 16 pregnant mice, we analyzed the frequency of B220^+^ B cells (using the gating strategy illustrated in Figure 1A), CD11b^−^Ly6G^+^ granulocytes and NOS2^+^CD11b^+^ macrophages (Figure 1B) in PL of WT, CD19^−/−^, BMyD88^−/−^ and µMT dams four hours after intraperitoneal LPS injection.

In WT dams, the main immune cell population in the peritoneal cavity (PC) were B cells, which were not impacted by LPS treatment (Figure 2A). The frequency of PC B cell in CD19^−/−^ dams were about half of that in the WT strain, yet, increased even further four hours after LPS injection. For BMyD88^−/−^ dams, the frequency of B cells declined following LPS application. As expected, µMT mice were B cell deficient (Figure 2A). Following LPS injection, all investigated mouse strains exhibited an increase in the percentage of CD11b^−^Ly6G^+^ granulocytes in the PC; however, this increase was more pronounced in WT mice than in CD19^−/−^, BMyD88^−/−^ and µMT dams (Figure 2B). While WT dams did not display an induction of NOS2 by CD11b^+^ peritoneal macrophages following LPS, the frequency of NOS2^+^CD11b^+^ cells was enhanced in CD19^−/−^ and BMyD88^−/−^ mice (Figure 2C).

LPS treatment induced the release of several chemokines in PL. The concentrations of TARC (Thymus and activation regulated chemokine; CCL17) and LIX were higher in WT than in CD19^−/−^, BMyD88^−/−^ and µMT dams (Figure 3A,B) four hours after LPS injection. The level of KC was increased following LPS in all investigated mouse strains and was lower in WT than in µMT dams after LPS treatment (Figure 3C), except for BMyD88^−/−^ mice. In CD19^−/−^, BMyD88^−/−^ and µMT mice, several chemokine levels such as MIP-1α (macrophage inflammatory protein-1α) and Eotaxin were already increased in PBS controls with a further increase observed after LPS application (Appendix A). 

The level of TARC in the serum of WT dams remained unchanged following LPS injection (Figure 3D). However, LIX and KC levels were increased in maternal serum following LPS treatment in WT, CD19^−/−^, BMyD88^−/−^ and µMT dams (Figure 3E,F). TARC, LIX and KC levels were significantly reduced in LPS-treated WT dams compared to in CD19^−/−^, BMyD88^−/−^ or µMT dams (Appendix A).

IL-6 expression was induced in all investigated strains following LPS injection in peritoneal lavage, with IL-10 release observed only in WT dams (Figure 4A,B). Among the strains treated with LPS, levels of IL-6 (Figure 4A), IL-10 (Figure 4B), TNF-α, and IL-17A (Appendix A) were higher in WT dams than in CD19^−/−^, BMyD88^−/−^ and µMT dams. In maternal serum, the levels of IL-6 (Figure 4C) and IL-10 (Figure 4D), TNF-α and IL-17A were increased four hours after LPS challenge in all investigated strains (Appendix A). There were no significant differences in IL-6 concentration within maternal serum among the LPS-treated groups (Figure 4C); however, WT LPS dams had significantly higher levels of IL-10 compared to CD19^−/−^, BMyD88^−/−^ and µMT dams (Figure 4D).

The data shows that B cell proficient WT dams respond to an immune-activating stimulus with a rapid and potent local inflammatory and regulatory immune response, in contrast to dams that are either B cell deficient or lacking the B cell-specific expression of CD19 or MyD88.

### 2.2. The LPS-Induced Inflammatory Immune Response Is Attenuated in WT Dams in Placenta, Amniotic Fluid and Fetal Serum Compared to Dams Lacking B Cells or B-Cell-Specific CD19 or MyD88 Expression 

After being removed four hours after LPS injection, placentae were cultured as an explant for 24 h, and then their supernatants were harvested and analyzed. We found that the expression of MCP-1, IL-12p70, IL-27 and IL-17A was induced in all mouse strains following LPS treatment. Moreover, we observed that the release of IL-1α was lower in placentae from LPS-treated WT dams than in those taken from CD19^−/−^, BMyD88^−/−^ and µMT dams (Appendix A). This same trend was also observed for IL-6 (Figure 5A) and TNF-α (Appendix A). Placentae from LPS-treated WT dams released higher IL-10 concentrations compared to those from CD19^−/−^, BMyD88^−/−^ and µMT dams (Figure 5B).

In the amniotic fluid (AF), the levels of IL-6 (Figure 5C) and TNF-α (Appendix A) were lower, while the concentration of IL-10 (Figure 5D) was higher in WT dams than in CD19^−/−^, BMyD88^−/−^ and µMT dams. 

The concentration of IL-6 in fetal serum was lower in fetuses obtained from WT compared to CD19^−/−^, BMyD88^−/−^ as well as µMT dams four hours after LPS injection (Figure 5E). Additionally, the level of TNF-α (Appendix A) was lower in LPS-treated WT dams than CD19^−/−^ mice. Following LPS treatment, there was an increase in IL-10 in fetal serum of WT mice compared to CD19^−/−^, BMyD88^−/−^ and µMT dams (Figure 5F).

These data show that B cells are involved in modulation in maternal immune response, resulting in reduced levels of inflammatory mediators in fetal compartments.

### 2.3. LPS Administration Altered the Expression of Inflammatory Mediators in Gestational and Fetal Tissues

The expression of several pro- and anti-inflammatory mediators (*Tnfa*, *Il10*, *Ifng*, *Il6*, *Mmp9*, *Il1b* and *Ccl5;* the list of primers is shown in Appendix A) was analyzed via PCR in gestational (placenta, amnion, uterus) and fetal tissues (liver, lung, gut, brain). The expression of *Il6* and *Ccl5* increased after LPS challenge in all investigated mouse strains in uterus, amnion and placenta (Figure 6A–C), with no statistically significant differences among the LPS-treated groups. An exception is the decreased expression of *Il6* in the placenta of LPS-treated WT dams compared to the placental *Il6* expression of µMT mice (Figure 6C). Furthermore, the expression of *Il10* in the uterus was higher in LPS-treated WT than BMyD88^−/−^ dams (Appendix A). There were no statistical significant differences found in the LPS-induced *Ccl5* expression between the investigated mouse strains in the uterus (Figure 6D), amnion (Figure 6E), and placenta (Figure 6F). 

Furthermore, the expression of *Tnfa* and *Il1b* was induced in all mouse strains in uterine tissue, amnion and placenta following LPS injection (Appendix A). The expression of *Ifng* in uterus increased four hours after LPS only in CD19^−/−^, BMyD88^−/−^ and µMT dams. In amnion, the *Ifng* expression increased after LPS injection in WT, CD19^−/−^ and BMyD88^−/−^ dams. LPS-induced *MMP-9* expression was detectable in the placenta of all mouse strains except for WT mice. There were no statistically significant differences detected within the LPS groups (Appendix A). 

*The Il10* expression could not be detected in fetal tissues. Expression of some mediators, such as *Il6*, *Il1b*, *Tnf*, *Ifng* and *Ccl5,* was changed in tissues like the fetal liver, lung and gut four hours after LPS administration. Fetuses gained from LPS-treated CD19^−/−^ dams exhibited a significantly higher expression of *Ifng* and *Mmp9* in the liver, *Il1b* in the lung and a reduced expression of *Il6* in the brain as compared to fetuses of the LPS-treated WT dams. Fetuses gained from the LPS-treated BMyD88^−/−^ and µMT mice showed reduced *Ccl5* expression in the gut. Fetuses from µMT mice exhibited higher expression of *Ifng* in the liver as well as *Il1b and Ccl5* in the lung as compared to the fetuses taken from WT dams. The expression of all examined molecules was low in fetal brain, with notable differences observed in the expression of *Il6* (Appendix A).

## 3. Discussion

PTB is a multifactorial pregnancy complication. Alterations in the intricate interplay between the immune system, reproductive tissues and gestational hormones may account for a part of preterm deliveries. Our previous research revealed a reduction of the serum levels of progesterone, a crucial pregnancy hormone known to suppress uterine contractions, 24 h after a LPS challenge in CD19^−/−^, BMyD88^−/−^ and µMT dams, which ultimately delivered [17]. Several hormone functions are mediated by the progesterone induced blocking factor (PIBF). B cells are a significant source of PIBF late in pregnancy in human and murine gestational tissues. IL-33 induces the expression of PIBF [19,20]. IL-33 and PIBF are involved in the protection against PTB [19]. Additionally, B cells may contribute to preeclampsia, a hypertensive pregnancy disorder also associated with an activation of the maternal inflammatory immune response [21]. Furthermore, we have shown that B cells play a role in the pathophysiology of PTB. B cells upregulated molecules involved in the pattern recognition and antigen presentation and exhibited a transition from the immune regulatory towards inflammatory phenotypes in patients with PTB [12,13,14,22]. Previous research suggests that B1 cells and plasmablasts in amniotic fluid, which have both the capacity to produce IL-10, were affected in women experiencing PTB and chronic chorioamnionitis, but not in cases of preterm prelabor rupture of membranes [23,24]. Through our studies in mice, we discovered the significance of the B cell-specific MyD88 and IL-10 expression for the appropriate fetal growth, as well as protecting against inflammation-induced PTB and the maternal well-being after LPS-induced PTB [15,17,18]. Here, we investigated the early immune reactions in full competent B cell-bearing WT dams that provide protection against PTB and compared them to those in the mice strains with either lacking or impaired B cells.

Our findings suggest that shortly after LPS challenge, fully functional B cell competent WT mice limit the inflammatory immune response in the gestational tissues including amnion and placenta. This results in a reduced inflammation in the fetal tissues. Contrarily, in the B cell-modified strains, inflammation is detected early in the gestational and fetal tissues, ultimately leading to PTB. Four hours after a LPS injection, a rapid recruitment of granulocytes without NOS2 induction by macrophages was detected in the PC of the WT dams. Additionally, we ascertained an expeditious induction of cytokines and chemokines, such as IL-6, TNF-α, and IL-10, along with chemokines TARC and MCP-1, which constrain the inflammatory immune response within the maternal compartment, consequently reducing the threat to the fetuses. 

The significance of the fully competent LPS signaling for the early recruitment of granulocytes and macrophages to the peritoneal cavity was established in a neonatal mouse model. TRIF2^−/−^ neonates, injected with *E. coli,* exhibited early elevation in inflammatory mediators, like IL-6, IL-1β, TNF-α, KC and MIP-1α, effectively restraining systematic infection [25]. This study presents evidence that B cells contribute to the initial phase of the host defense. WT B cells, which express all cell-specific molecules necessary for the LPS response, exhibit an early recruitment of innate immune cells to the PC and a robust release of inflammatory mediators. The disruption of this process may occur due to B cell deficiency and loss of B cell-specific MyD88 or CD19 expression.

B cells comprise the major immune cell population in the PC of WT dams. Conversely, CD19^−/−^ dams exhibited lowered B cell numbers. The significance of CD19 in the B1 cell development is established [26], although we detected a rise in peritoneal B cell numbers induced by LPS despite the CD19 loss. Further investigation is required to ascertain, whether this was due to a compensatory mechanism mediated by the TLR4/ MyD88 signaling pathway following the loss of the RP105/ CD19-mediated LPS signaling. Peritoneal B1 cells are the primary source of antibodies aimed at T cell-independent antigens, including LPS. B cells produce antibodies that efficiently activate granulocytes [27,28], while granulocytes produce cytokines necessary for the survival, maturation and differentiation of B cells [29,30,31]. This interaction may also be critical for the LPS-induced PTB.

Various cell types including trophoblasts, T cells and macrophages produce NOS2, also referred as inducible nitric oxide synthase 2. NOS2 production can be induced by LPS challenge [32] and is upregulated before and particularly during labor. We also demonstrated an upregulation of the enzyme in our mouse model of LPS-induced PTB [33,34]. Nitric oxide donors such as nitroglycerin can be effective as acute tocolysis in patients with threatened PTB [35]. NO produced by NOS2 can restrict the differentiation of M1 and Th17 cells, which may constrain the protective immune response to inflammatory stimuli [36,37]. This mechanism could also be relevant in our model, as the IL-17A levels in the PL and serum of WT dams were higher as compared to all other investigated strains. We observed a significant NOS2 induction by macrophages in mice with B cell deficiency (µMT) and with lacking B cell-specific expression of MyD88 (BMyD88^−/−^) or CD19 (CD19^−/−^). Consequently, our data would imply that WT mice with fully competent B cells restrict NOS2 expression by macrophages facilitating a robust inflammatory immune response in the PC to control inflammation and, therefore, capable to limit the inflammation to the maternal compartment.

TARC is expressed during pregnancy facilitating migration and infiltration of T cells into the feto-maternal interface and, thereby, ensuring the maintenance of pregnancy [38]. Women delivering at term demonstrate a higher TARC expression as compared to those, who were diagnosed with PTB [39]. In the present study, we observed a robust TARC upregulation in the PL of WT mice, which can be interpreted as a proactive response to safeguard pregnancy against inflammation. Further, the LPS challenge induces secretion of MCP-1 that shifts a balance between Th1 and Th2 cells resulting in an increased secretion of IL-4 and IL-10 and enhanced CD80 and CD86 expression at the feto-maternal interface [40]. LPS can also directly activate peritoneal B cells, stimulating Breg cell properties such as IL-10 and IL-6 secretion [41]. All these mechanisms may play a role in the B cell-dependent protection against PTB triggered by inflammation [18]. 

A number of chemokines, specifically KC, MDC and MCP-1, were reduced in the maternal serum of WT dams four hours after LPS injection, in comparison to the CD19^−/−^, BMyD88^−/−^ and µMT dams. In contrast, the serum levels of IL-10 were higher in WT dams than in other mouse strains that were investigated. The significance of swift IL-10 secretion by B1 cells as a counterbalance to the release of pro-inflammatory mediators after lethal LPS injection was established using B1-deficient Xid mice [42]. Restriction of the inflammation to a local response with limited systemic inflammatory reaction is likely to facilitate the protection against LPS-induced PTB in WT mice. Indeed, in contrast to CD19^−/−^, BMyD88^−/−^ and µMT dams, we observed an enhanced release of IL-10 by B cells and expression of CD4^+^ and CD8^+^ by T cells in blood of WT dams 24 h after LPS exposure [17]. In the present work, we provide evidence that the imbalance between inflammation and immune-regulation occurs even much earlier as that and can be detected already four hours after LPS application. 

LPS-induced PTB was associated with elevated levels of TNF-α and IL-6 in the maternal serum, as well as with increased TNF-α levels in the fetal serum and enhanced TNF-α and IL-6 concentrations in the amniotic fluid [43]. Our study confirmed the increased IL-6 and TNF-α level in maternal serum of WT dams after LPS challenge. However, both these cytokines were lower in the amniotic fluid and fetal serum in WT mice in comparison to CD19^−/−^, BMyD88^−/−^ and µMT mice. In line with this, higher levels of TNF-α and IL-6 in the mother as compared to the fetus were observed by Hudalla et al. after LPS challenge at gd 17, even though the dose was too low to cause labor [44]. This implies that the rapid conversion to immune regulation in the mother protects the fetuses from extensive inflammatory mediators. 

Bommer at al. reported on an increase of B cells in the amniotic fluid following LPS injection and suggested that B cells may serve as an immunological protective barrier against PTB [45]. Earlier studies found that administration of high-dose LPS i.p. led to the expression of *Il1b* and *Tnfa*, but not *Il6* or *Il10*, in the placenta [46]. We also observed lower uterine, amniotic and placental expressions of *Il6* and *Ccl5* in WT mice as compared to CD19^−/−^, BMyD88^−/−^ and µMT strains. In particular, B cell deficiency appears to have a detrimental effect on the expression of *Il6* and *Ccl5* in the placenta, indicating a significant role of B cells in the controlling the inflammatory immune response and recruiting of leukocytes towards the feto-maternal interface. Tulina et al. [47] demonstrated a release of IL-6, IL-1β and RANTES in amniotic fluid and increased *Il6*, *IL1b and Ccl5* expression in the placenta after intra-amniotic administration of LPS on the gd 15 in WT mice. In contrast, IL-6 and IL-1β release and *Il6* and *Il1b* expression remained unaltered in genome-wide MyD88^−/−^ mice, whereas the TRIF-dependent *Ccl5* expression in the placenta and RANTES levels in the amniotic fluid were even higher as compared to WT mice [47]. We also observed an increase of *Ccl5* expression, particularly in the amnion of BMyD88^−/−^ dams. This underscores the contribution of the B cell-specific MyD88^−/−^ signaling to immune response on a LPS challenge during pregnancy, ultimately determining the likelihood of preterm delivery.

No major changes were found in fetal tissues, likely due to the early stage. Some studies have reported moderate increases in the expression of *Il1b*, *Il6* and *Tnfa* in the fetal lung after i.p. LPS injection at five hours, while no detectable increase in *Il10* expression was found [46]. Additionally, no significant changes in the expression of *Il1b*, *Mip2*, *Tnfa* and *Il6* were observed in fetal organs at three hours or eight hours after maternal LPS exposure [43]. To investigate the contribution of B cell deficiency or B cell-specific molecules involved in LPS signaling to LPS-induced PTB in fetal compartments, selecting later time points is necessary.

## 4. Materials and Methods

### 4.1. Animals and Mouse Model

C57BL/6 (H2^b^) wildtype (WT) littermates, CD19^cre/wt^ MyD88^flox/flox^ (B6.129P2(SJL)-*Myd88^tm1Defr^*/J) B-cell-specific MyD88-deficient (BMyD88^−/−^) mice, CD19^−/−^ mice lacking CD19 expression (B6.129P2(C)^Cd19tm1(cre)Cgn^/J) and B-cell-deficient µMT mice (B6.129S2-*Ighm^tm1Cgn^*/J) were bred at the animal facility of the Medical University of Magdeburg. BALB/c (H2^d^) males were purchased from Janvier (Le Genest-Saint-Isle, France). All of the mice were kept in open cages in our animal facility in a 12 h light cycle under optimal conditions. Chow and water were applied ad libitum. Animal experiments complied with the ARRIVE guidelines and were carried out according to institutional guidelines after ministerial approval and in conformity with the European Communities Council Directive (EU Directive 2010/63/EU for animal experiments; approval number: 42502-2-1332 Uni MD).

Virgin female WT, CD19^−/−^, BMyD88^−/−^ and µMT mice aged 8–12 weeks were mated with BALB/c males. Females were inspected twice a day for vaginal plugs. The presence of a vaginal plug was designated as day 0 of pregnancy (gestational day, gd 0).

### 4.2. The Application of LPSs and Sample Collection

A sample of 200 µL LPSs (E. coli serotype 0111:B4; Sigma Aldrich, Taufkirchen, Germany) at a concentration of 0.4 mg/kg body weight (BW) and diluted in PBS was injected intraperitoneally at gd 16, as previously described [18]. The mice were sacrificed via cervical dislocation 4 h later. Peritoneal lavage (PL) was carried out by flushing the peritoneal cavity with 1 mL of 0.9% NaCl solution (Fresenius Kabi, Bad Homburg, Germany). The sample was centrifuged at 1200× *g* rpm, and the supernatant was collected and stored at −80 °C. The cell pellet was suspended in FACS buffer and stained (see below). Blood was obtained via puncture of the heart and stored in heparinized tubes on ice. Maternal and fetal blood was centrifuged at 7000× *g* rpm for 10 min at room temperature (RT). Serum was collected and stored at −80 °C. Uterus, placenta, amnion and fetal tissues (lung, liver, gut and brain) were collected, snap-frozen and stored at −80 °C. 

### 4.3. Cell Staining and Flow Cytometry

Single-cell suspensions from peritoneal lavage were obtained by flushing into the peritoneal cavity with 5 mL ice-cold 1% FBS/PBS as recently described [15] and analyzed using an Attune NxT flow cytometer (Thermo Fisher Scientific, Dreiech, Germany). 

Cells were stained with PerCP-Cy5.5-labeled anti-CD45, FITC-labeled anti-Ly6G, PE-labeled anti-NOS2 and APC-labeled anti-CD11b or FITC-labeled anti-CD45 and PerCP-Cy5.5-labeled anti-B220 antibodies (all BD Biosciences, Heidelberg, Germany) for 30 min at 4 °C. Gates were set using FMO controls. Measurements were performed on an Attune NxT flow cytometer (Thermo Fisher Scientific, Dreiech, Germany). Data were analyzed using FlowJo software 10.8.0. 

### 4.4. Cytokine Detection in Sera and Supernatants

Cytokines were measured using the 13-plex cytometric bead array (CBA) mouse Th1/Th2/Th17 Cytokine Kit from BD Biosciences (Heidelberg, Germany) and 13-plex Legendplex Mouse Inflammation Panel (Biolegend, San Diego, CA, USA). Chemokines were determined with the 13-plex Legendplex Mouse Proinflammatory Chemokine Panel (Biolegend, San Diego, CA, USA) following the supplier’s recommendation. 

### 4.5. Real-Time Reverse Transcriptase Polymerase Chain Reaction (RT-PCR)

Total RNA from frozen tissues was isolated using TRIzol (Ambion, Thermo Fisher Scientific, Dreieich, Germany) and a homogenizer (Ultra Turrax T8; Ika, Staufen, Germany). RNA was extracted with chloroform, precipitated with isopropanol, washed in 80% ethanol and diluted in RNase-free water (Braun, Melsungen, Germany). RNA quantity and quality was determined using an NP80 NanoPhotometer (Implen, Munich, Germany). For cDNA synthesis, 2 µg total RNA was incubated with oligo dTs for 10 min at 75 °C and 5 min on ice. Subtracted mRNA was incubated with dNTPs (2.5 mmol/l), DNase I (1 U/mL) and RNase inhibitor (40 U/mL) for 30 min at 37 °C. Afterwards, it was heated for 5 min at 75 °C. Reverse transcriptase (200 U/mL) and RNase inhibitor (all reagents from Promega) were added, followed by incubation for 60 min at 42 °C and for 5 min at 94 °C.

RT-PCR amplifications were performed using an iCycler (iQ5, BioRad, Feldkirchen, Germany). Primers for *Il1b*, *Il6*, *Il10*, *Tnfa*, *Ccl5* (RANTES), *Mmp9* and *Actb* (β actin) are listed in Appendix A. Experiments were run in duplicate with an initial denaturation for 5 min at 95 °C, followed by 40 cycles of denaturation for 45 s at 95 °C and annealing for 60 s at 60 °C for amplification.

### 4.6. Data Analysis and Statistics

Statistical analysis was performed using GraphPad Prism 8.0 software. Normality of distribution was determined with the Shapiro–Wilk test. Depending on the result, data were analyzed either using one-way ANOVA followed by the Holm–Sidak multiple comparisons test or Kruskal–Wallis test, followed by Dunn’s multiple comparisons test. Significance was defined as follows: * *p* < 0.05, ** *p* < 0.01, *** *p* < 0.001, **** *p* < 0.0001.

## 5. Conclusions

Overall, our study has demonstrated the impact of the early B cell response to an inflammatory stimulus for the further course of pregnancy, particularly in preventing PTB. B cells may play a key role at an early stage of inflammation by interacting with other immune cells and/ or releasing mediators. In this process, the B cell-specific expression of proteins involved in the LPS signaling, such as MyD88 and CD19, seem to be crucial. Both molecules appear to regulate the local immune response in the PC, thereby limiting the spread of inflammation towards gestational and fetal tissues.

## Figures and Tables

**Figure 1 ijms-24-16091-f001:**
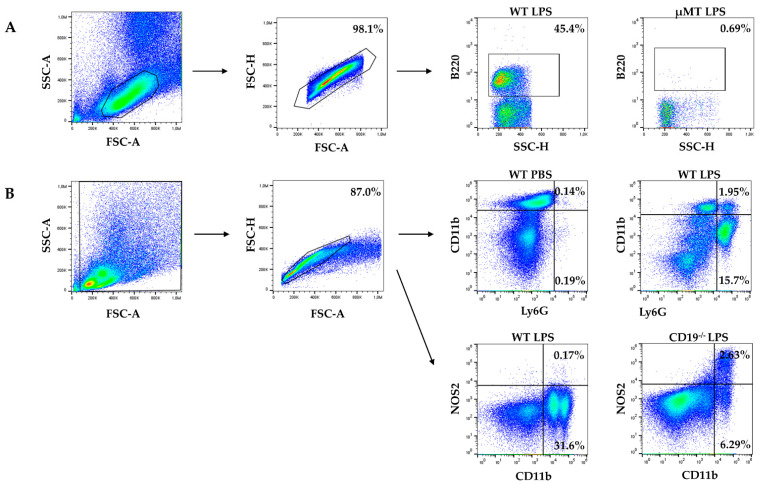
Gating strategy for flow cytometry of peritoneal immune cells. (**A**) The percentages of B220^+^ B cells were determined by setting the lymphocyte gate, followed by single cell gating and finally B220^+^ gating. (**B**) The percentages of CD11b^−^LyG6^+^ granulocytes and NOS2^+^CD11b^+^ macrophages were determined by setting a cell gate, followed by single cell gating and finally detection of cells stained with anti-Ly6G and anti-CD11b or anti-CD11b and anti-NOS2. Representative pictures of two different mouse strains 4 h after PBS or LPS injection are presented.

**Figure 2 ijms-24-16091-f002:**
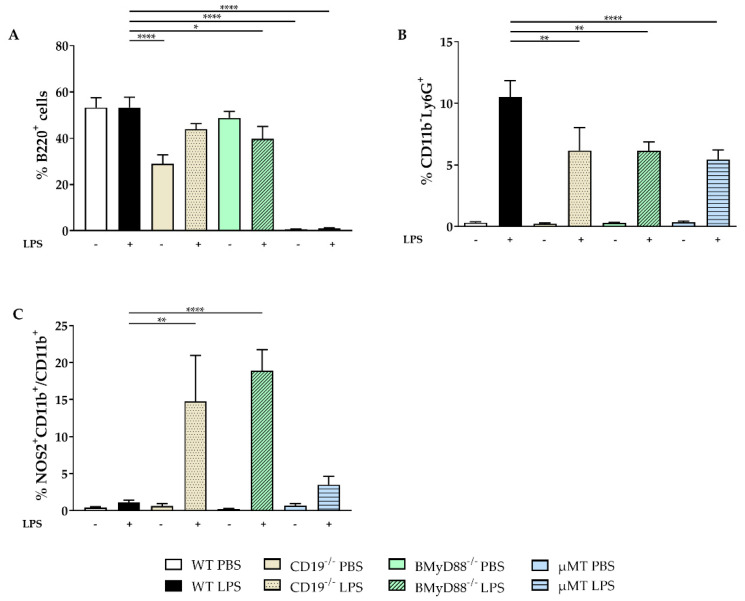
LPS-induced alterations in peritoneal immune cells. Four hours after PBS or LPS challenge, the immune cell populations within the peritoneal cavity of WT, CD19^−/−^, BMyD88^−/−^ and µMT dams were examined by flow cytometry. Following the Shapiro–Wilk test, data were analyzed with either one-way ANOVA, followed by the Holm–Sidak multiple comparisons test, or using the Kruskal–Wallis test, followed by Dunn’s multiple comparisons test (with the WT LPS group as a comparison group). The percentages of B220^+^ B cells (**A**), CD11b^−^LyG6^+^ (**B**) and NOS2-expressing CD11b^+^ macrophages (**C**) are presented as mean ± SD. n = 5–6 mice/group. * *p* < 0.05, ** *p* < 0.01, **** *p* < 0.0001.

**Figure 3 ijms-24-16091-f003:**
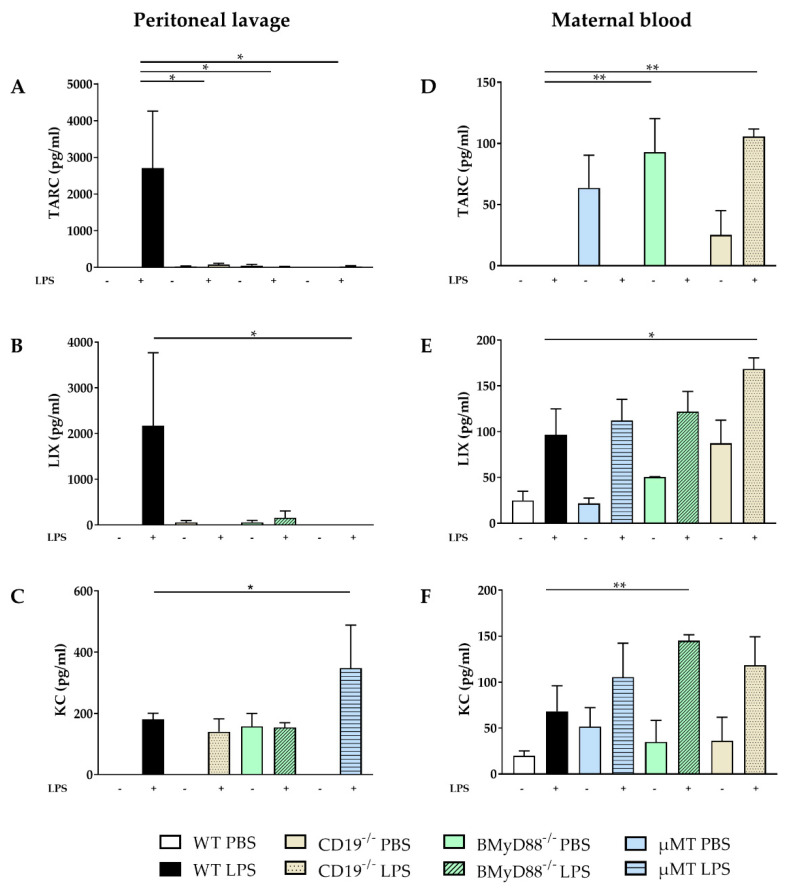
LPS-induced changes in the expression of TARC, LIX and KC in maternal compartments. The levels of TARC, LIX and KC were determined by cytometric bead arrays in peritoneal lavage (**A**–**C**) and maternal serum (**D**–**F**) 4 h after i.p. PBS or LPS injection in WT, CD19^−/−^, BMyD88^−/−^ and µMT dams. Following the Shapiro–Wilk test, data were analyzed either with one-way ANOVA, followed by the Holm–Sidak multiple comparisons test, or using the Kruskal–Wallis test, followed by Dunn´s multiple comparisons test (with the WT LPS group as a comparison group). Data are presented as mean ± SD. n = 5–6 mice/group. * *p* < 0.05, ** *p* < 0.01.

**Figure 4 ijms-24-16091-f004:**
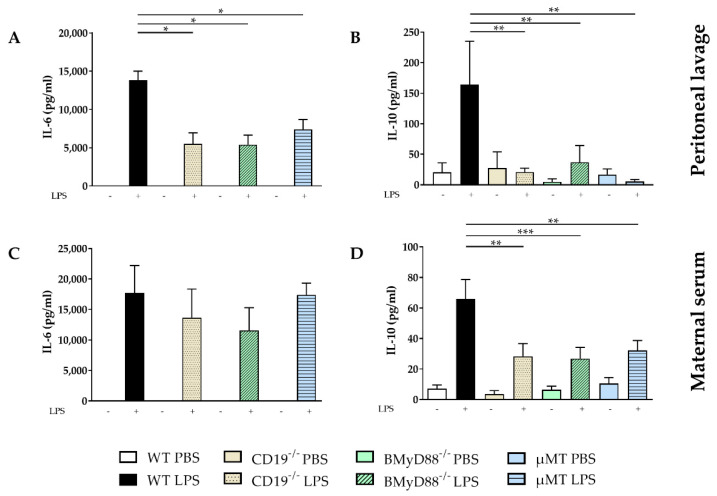
LPS-induced changes in the expression of IL-6 and IL-10 in maternal compartments. The levels of IL-6 and IL-10 were determined by cytometric bead arrays in peritoneal lavage (**A**,**B**) and maternal serum (**C**,**D**) 4 h after i.p. PBS or LPS injection in WT, CD19^−/−^, BMyD88^−/−^ and µMT dams. Following the Shapiro–Wilk test, data were analyzed with one-way ANOVA, followed by the Holm–Sidak multiple comparisons test, or using the Kruskal–Wallis test, followed by Dunn´s multiple comparisons test (with the WT LPS group as a comparison group). Data are presented as mean ± SD. n = 5–6 mice/group. * *p* < 0.05, ** *p* < 0.01, *** *p* < 0.001.

**Figure 5 ijms-24-16091-f005:**
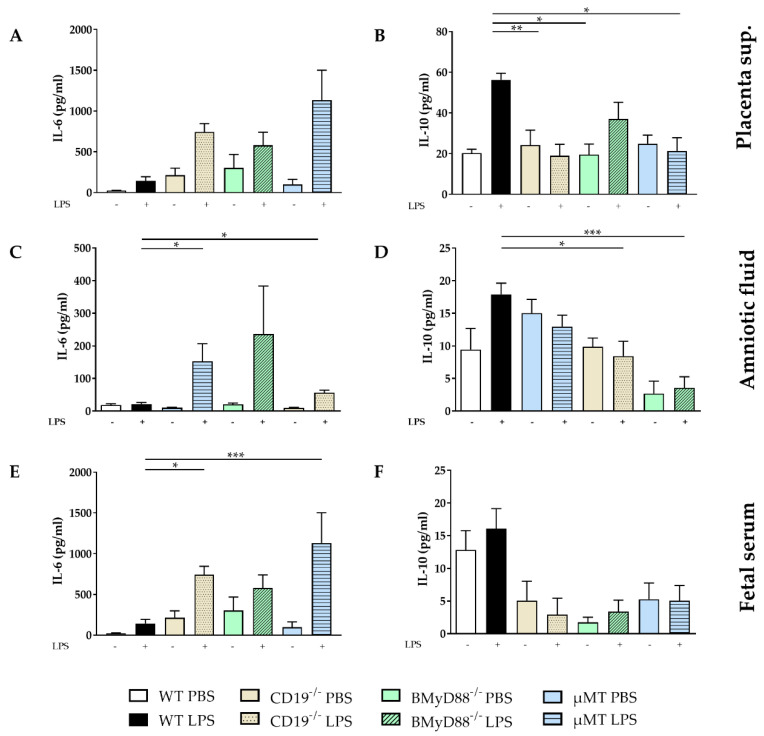
LPS-induced changes in the expression of IL-6 and IL-10 in fetal compartments. The levels of IL-6 and IL-10 were determined via cytometric bead arrays in placental supernatant (**A**,**B**), amniotic fluid (**C**,**D**) and fetal serum (**E**,**F**) 4 h after i.p. PBS or LPS injection in WT, CD19^−/−^, BMyD88^−/−^ and µMT dams. Following the Shapiro–Wilk test, data were analyzed with one-way ANOVA, followed by the Holm–Sidak multiple comparisons test, or using the Kruskal–Wallis test, followed by Dunn´s multiple comparisons test (with the WT LPS group as a comparison group). Data are presented as mean ± SD. n = 5–6 mice/group. * *p* < 0.05, ** *p* < 0.01, *** *p* < 0.001.

**Figure 6 ijms-24-16091-f006:**
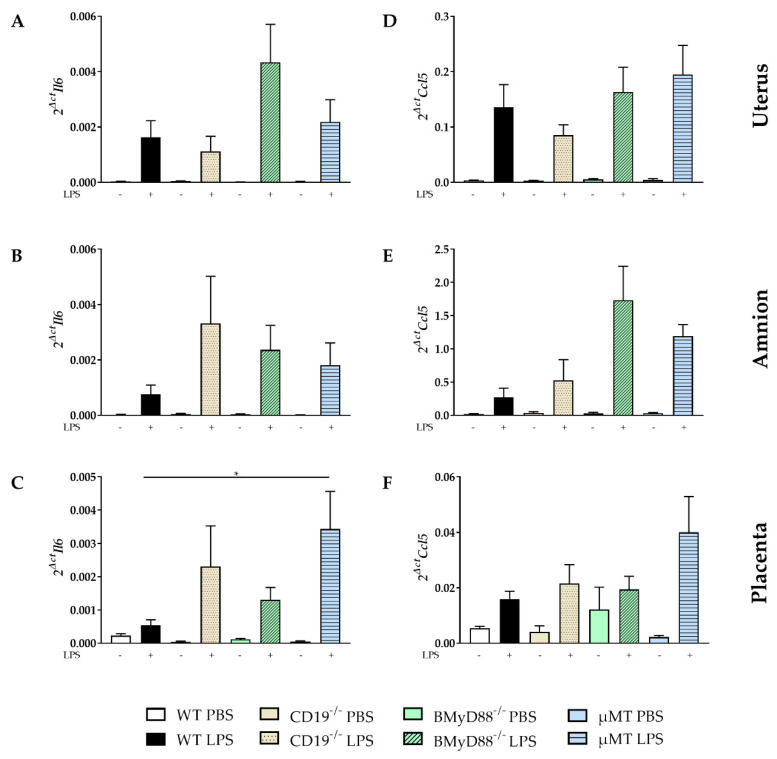
The LPS-induced expression of *Il6* and *Ccl5* is influenced by the expression of maternal B-cell-specific molecules in gestational tissues. The *Il6* mRNA level (**A**–**C**) and the expression of *Ccl5* mRNA (**D**–**F**) were determined 4 h after PBS or LPS injection in the uterus (**A**,**D**), amnion (**B**,**E**) and placenta from WT, CD19^−/−^, BMyD88^−/−^ and µMT by quantitative real-time PCR. Data were analyzed using the Kruskal–Wallis test, followed by Dunn´s multiple comparisons test (with the WT LPS group as a comparison group) and presented as mean ± SD. n = 5–6 mice/group. * *p* < 0.05.

## Data Availability

The datasets in this study are available from the corresponding author upon reasonable request.

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
