# Peer review of "B Cells Induce Early-Onset Maternal Inflammation to Protect against LPS-Induced Fetal Rejection"

_ijms, 2023, doi:10.3390/ijms242216091_

Round 1
Reviewer 1 Report
Please find my comments in the attached file.

The quality of the english language needs to be improved by a native speaker. The way many sentences are written do not meet the scientific standards.
Author Response
Answers to the reviewer
We thank the Reviewer for the time and the dedication to review our manuscript. We highly appreciate the suggestions and the constructive criticisms of reviewer #1 that helped us to improve our manuscript. Please find attached a point-by-point reply to reviewer #1 comments. The changes are marked in red in the revised manuscript file.
We apologize for the poor English grammar and writing style and hope to have improved both by making extensive changes to the manuscript.
Major corrections
Title/Abstract
We agree and changed the title: B cells induce early-onset maternal inflammation to protect against LPS-induced fetal rejection”.
Last sentence: LPS stimulates full-competent WT B cells to differentiate into Breg cells that was shown by our group and by several other groups. Breg cells are protective in pregnancy and a decrease of Breg cells were associated with human and murine pregnancy complications. If the immune system has to deal with LPS, induction of Breg cells is a positive effect that prevents fetal rejection. However, we rewrote the sentence.
Methods
All mice were kept in one room in open cages. “Optimal conditions” refer to the 12h light-dark cycle, a constant temperature between 21°C and 22°C and 55% humidity. All groups were kept equally.
Figures/tables
We decided against the use of dots since we determined that due to the number of groups, it´s even more confusing. Instead, we decided to use different colors to improve clarity.
We apologize for the mistake in Figure 1B. We corrected it.
In order to improve clarity, we present the most important results as Figures and put all tables in the supplement.
We apologize for not presenting LIX in PC. We corrected it.
Normality of distribution was determined by Shapiro-Wilk test. Each group was tested individually. If one group did not pass the normality test (α=0.05), analysis was performed by Kruskal-Wallis test, otherwise One-way ANOVA. Afterwards, the WT LPS group was compared to each other group. These calculations were performed either by Dunn´s multiple comparisons test or by Holm-Sidak multiple comparisons test. Each marker in each biological material was tested separately. The p value shows the result of the One-way ANOVA or Kruskal-Wallis test.
We did not put any other values such as SD into the table since it would double the numbers and decrease clarity a lot.
We used the appropriate statistical method for calculation of our data. Indeed, we are sure that the mathematical results concerning normality distribution are correct.
We added the gating strategies in Figure S1. Gates were set using FMO controls.
Results
We used generalization concerning the group WT LPS since this is our reference group. Of course, other groups also showed differences among each other, but among the LPS groups, only WT dams did not deliver preterm and the meaning of our study was to investigate differences between WT and the B cell modified groups to identify marker(s) that might explain the outcome.
Lines 151-152: We agree and we added the following sentence: “The level of KC was increased following LPS in all investigated mouse strains”.
Lines 152-154: We agree with the reviewer. We added the PBS groups from the other strains.
Lines 194-196: TNF-α was mentioned in placenta supernatant, amniotic fluid and fetal serum.
Lines 194-196: “No differences” mean indeed “no statistically significant differences”. We apologize and we corrected this mistake.
Line 220: We apologize for this mistake and added more information.
Discussion
We would like to thank the reviewer for bringing these publications to our attention. We have added them to the discussion.
Minor corrections/comments
Abstract
We added the word “murine” at an earlier point in the abstract. We rewrote the last sentence of the abstract.
Introduction
Line 40: We added “MyD88-dependent” signaling pathways.
Methods
We added “200µl LPS (E. coli serotype 0111:B4; Sigma Aldrich, Taufkirchen, Germany) at a concentration of 0.4 mg/kg body weight (BW) diluted in PBS.”
Mice were sacrificed by “cervical dislocation” (added to Methods part).
We added the information that the cells were obtained “by flushing to peritoneal cavity with 5ml ice-cold 1% FBS/PBS”.
Results
Line 139: We apologize for this misunderstanding and rewrote: “WT dams induce a robust immune response detectable in peritoneal lavage (PL) and maternal serum 4h after LPS challenge”.
fine-tuning: We thank the reviewer for these hints and corrected them.
We described some results in more detail in the discussion.
Paragraph 213-222: We rewrote the second sentence for clarity.
We apologize for the linguistic mistakes. We examined the manuscript thoroughly for linguistic errors.
We added a space between numbers and units. We wrote the phenotypes in superscript.
We added the missing abbreviation descriptions (TIR, RT).
We apologize for the inconsistencies. We corrected them.
Section 3.3: We added the method (“The expression of several pro- and anti-inflammatory mediators (TNF-α, IL-10, IFN-γ, IL-6, MMP9, IL-1β and RANTES) were determined in gestational (placenta, amnion, uterus) and fetal tissues (liver, lung, gut, brain) by PCR.”).
Tables: It is correct that values under the detection limit were handled as zero values, also in terms of statistical analysis.
Below are the data from the peritoneal cell counts. It is obvious that WT and BMyD88-/- dams increase the cell number in the PC while CD19-/- and µMT mice showed a decreased cell count. Since we actually investigate the reasons for this different result (proliferation, migration, apoptosis) we kindly ask you to treat this data confidentially.
Throughout the manuscript, we always presented percentages. The decrease in B cells in µMT mice is associated with an increase in T cells (CD4+ as well as CD8+).

Reviewer 2 Report
In the present work, Uehre et al. try to explain that B cells induce early-onset maternal inflammation to protect against fetal rejection. There are some questions that should be explained.
1. The manuscript is dotted with poor English grammar and writing style throughout the manuscript.
For example,
Line 18, ‘4h later, LPS-treated’; Line 68, ‘8-12 weeks old’……. In general, a sentence does not begin in arabic numeral. Please check it throughout the manuscript.
Lines 31-32, ‘The immune system in pregnancy has to protect mother and unborn against the consequences of microbial exposure.’ This sentence should be rewritten.
2. Line 91, ‘2.3. Cell staining and flow cytometry’. Cell staining and flow cytometry were performed. However, the representative pictures were not present in the Result section.
3. ‘min.’ and ‘sec.’ should be revised.
4. Figure 1 and 2, the * should be enlarged.
5. Line 146, ‘N=5-6 mice/group.’ ‘N’ should be in lower-case character. Please check it throughout the manuscript.
6. Gene abbreviations should be italicized.
7. Lines 229-232, this paragraph may be present in Discussion section.
8. A reference may be related to this topic.
Deer E, Herrock O, Campbell N, Cornelius D, Fitzgerald S, Amaral LM, LaMarca B. The role of immune cells and mediators in preeclampsia. Nat Rev Nephrol. 2023;19(4):257-270. doi: 10.1038/s41581-022-00670-0.
English very difficult to understand/incomprehensible.
Author Response
Answers to the reviewer
We thank the Reviewer for the time and the dedication to review our manuscript. We highly appreciate the suggestions and the constructive criticisms of reviewer #2 that helped us to improve the manuscript. Please find attached a point-by-point reply to reviewer #2 comments. The changes are marked in red in the revised manuscript file.
We apologize for the poor English grammar and writing style and hope to have improved both by making extensive changes to the manuscript.
- We checked the usage of Arabic numbers at the beginning and and have changed the sentences accordingly.
Lines 31-32: We rewrote the sentence.
- We added representative picture as Figure S1.
- We revised both abbreviations.
- We enlarged both figures.
- We corrected it and wrote “N” in lower-case character.
- We correct this mistake. We wrote the gene abbreviations in italics.
- Lines 229-232: We moved the paragraph to the Discussion.
- We would like to thank the reviewer for bringing this publication to our attention. We have added it to the discussion.
Round 2
Reviewer 1 Report
My comments can be found in the attached file.

The language has been improved as compared to the first version of the manuscript.
Author Response
We thank the Reviewer again for the time to review our manuscript and for reading the manuscript carefully. Please find attached a point-by-point reply to the comments. The changes are marked in red in the revised manuscript file.
Moderate:
Tables: We thank the reviewer for this suggestion. We placed an asterisk next to each value to mark statistically significant differences to the WT LPS group.
Figure 4b (now Figure 5b): We apologize for this mistake and removed the significance line and asterisk from the WT PBS group.
Results 3.3: We added that we also did not find statistical significant differences in Ccl5 expression in amnion and placenta. We added that Il6 was significantly increased in placenta of µMT dams compared to WT dams 4 h after LPS challenge and Il10 in uterus of WT LPS dams compared to BMyD88-/- dams. We agree that the sentence (242-245) is too general or biased. We mentioned only the significant values.
Figure 5: We exchanged the word “increased” with “altered”. We described only the significant values for the fetal tissues.
Minor:
We added the information that we used WT LPS as controls to the figure legends
We apologize that in Figure 4 asterisks had a different size. We corrected it.
Line 92: We corrected “into”.
Lines 95-96: We corrected the description of the antibodies and added the word “anti”.
Lines 150-152: We corrected the sentence.
Line 153: We corrected the sentence.
Line 167: We corrected the sentence.
Lines 172-175: We rewrote the paragraph and focused on the comparison of the LPS-treated groups.
Line 194: We added the results obtained from placenta supernatant (Table S3).
Line 203: We corrected the sentence.
Line 253: We corrected the word.
Line 278: We corrected the word (E. coli).
We thank the reviewer for providing us the helpful link to the gene nomenclature. We have replaced the nomenclature accordingly.
Reviewer 2 Report
Thanks for author’s responses. However, this manuscript should be supported by a professional English language proofreading service, and the Figure S1 should be present in the manuscript.
Extensive editing of English language required.
Author Response
We thank the Reviewer again for the time to review our manuscript and for reading the manuscript carefully. The changes are marked in red in the revised manuscript file.
We tried to further improve the manuscript linguistically and grammatically.
The Figure S1 is now present in the manuscript (Figure 1).
Because the reviewer has no concrete suggestions regarding improvements in the experimental design, the description of the methods and the presentation of the results, we hope that the improvements we have made are sufficient to optimize the understanding and readability of the manuscript.
Round 3
Reviewer 2 Report
Thanks for author’s responses. However, some genes should be in both capitalized and italic.
Minor editing of English language required.